# COLD POSTERIORS THROUGH PAC-BAYES

## ABSTRACT

We investigate the cold posterior effect through the lens of PAC-Bayes general-ization bounds. We argue that in the non-asymptotic setting, when the number of training samples is (relatively) small, discussions of the cold posterior effect should take into account that approximate Bayesian inference does not readily provide guarantees of performance on out-of-sample data. Instead, out-of-sample error is better described through a generalization bound. In this context, we explore the connections of the ELBO objective from variational inference and the PAC-Bayes objectives. We note that, while the ELBO and PAC-Bayes objectives are similar, the latter objectives naturally contain a temperature parameter $\lambda$ which is not restricted to be $\lambda = 1$. For classification tasks, in the case of Laplace approximations to the posterior, we show how this PAC-Bayesian interpretation of the temperature parameter captures important aspects of the cold posterior effect.

## 1 INTRODUCTION

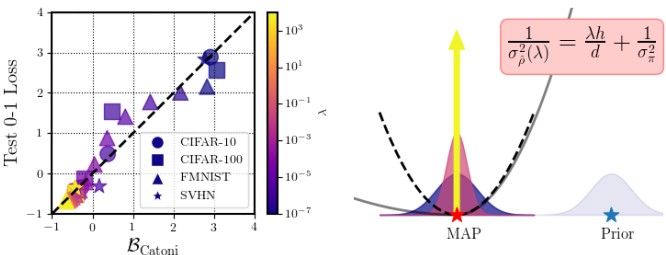

(a) Classification tasks      (b) Laplace, different temperatures $\lambda$

Figure 1: PAC-Bayes bounds correlate with the test 0-1 Loss for different values of the temperature $\lambda$ (quantities on both axes are normalized). (a) Classification tasks on CIFAR-10, CIFAR-100, and SVHN datasets ($\sigma_\pi^2 = 0.1$, ResNet22) and FMNIST dataset ($\sigma_\pi^2 = 0.1$, ConvNet). (b) Graphical representation of the Laplace approximation for different temperatures: for hot temperatures $\lambda \ll 1$, the posterior variance becomes equal to the prior variance; for $\lambda = 1$ the posterior variance is regularized according to the curvature $h$; for cold temperatures $\lambda \gg 1$, the posterior becomes a Dirac delta on the MAP estimate.

In their influential paper, Wenzel et al. (2020) highlighted the observation that Bayesian neural networks typically exhibit better test time predictive performance if the posterior distribution is "sharpened" through tempering. Their work has been influential primary because it serves as a well documented example of the potential drawbacks of the Bayesian approach to deep learning. While other subfields of deep learning have seen rapid adoption, and have had impact on real world problems, Bayesian deep learning has, to date, seen relatively limited practical use (Izmailov et al., 2021; Lotfi et al., 2022; Dusenberry et al., 2020; Wenzel et al., 2020). The "cold posterior effect", as the authors of Wenzel et al. (2020) named their observation, highlights an essential mismatch between Bayesian theory and practice. As the number of training samples increases, Bayesian theory tells states that the posterior distribution should be concentrating more and more on the true model parameters, in a frequentist sense. At any time, the posterior is our best guess at the true model parameters, without having to resort to heuristics. Since the original paper, a number of works (Noci et al., 2021; Zeno et al., 2020; Adlam et al., 2020; Nabarro et al., 2022; Fortuin et al., 2021;

Aitchison, 2021) have attempted to explain the cold posterior effect, identify its origins, propose remedies and defend Bayesian deep learning in the process.

The experimental setups where the cold posterior effect arises have, however, been hard to pinpoint precisely. Noci et al. (2021) conducted detailed experiments testing various hypotheses. The cold posterior effect was shown to arise from augmenting the data during optimization (data augmentation hypothesis), from selecting only the "easiest" data samples when constructing the dataset (data curation hypothesis), and from selecting a "bad" prior (prior misspecification hypothesis). Nabarro et al. (2022) propose a principled log-likelihood that incorporates data augmentation, however they show that the cold-posterior persists. Bachmann et al. (2022) also propose a mechanism by which data-augmentation leads to mispecification and how the tempered posterior alleviates it. They prove their results for simplified settings, and acknowledge that there might be other potential sources of the cold-posterior effect. Data curation was first proposed as an explanation in Aitchison (2021), however the author shows that data curation can only explain a part of the cold posterior effect. Misspecified priors have also been explored as a possible cause in several other works (Zeno et al., 2020; Adlam et al., 2020; Fortuin et al., 2021). Again the results have been mixed. In smaller models, data dependent priors seem to decrease the cold posterior effect while in larger models the effect increases (Fortuin et al., 2021).

We posit that discussions of the cold posterior effect should take into account that in the *non-asymptotic setting* (where the number of training data points is relatively small), Bayesian inference does not readily provide a guarantee for *performance on out-of-sample data*. Existing theorems describe *posterior contraction* (Ghosal et al., 2000; Blackwell & Dubins, 1962), however in practical settings, for a finite number of training steps and for finite training data, it is often difficult to *precisely* characterise how much the posterior concentrates. Furthermore, theorems on posterior contraction are somewhat unsatisfying in the supervised classification setting, in which the cold posterior effect is usually discussed. Ideally, one would want a theoretical analysis that links the posterior distribution to the *test* error directly.

Here, we investigate PAC-Bayes generalization bounds (McAllester, 1999; Catoni, 2007; Alquier et al., 2016; Dziugaite & Roy, 2017) as the model that governs performance on out-of-sample data. PAC-Bayes bounds describe the performance on out-of-sample data, through an application of the convex duality relation between measurable functions and probability measures. The convex duality relationship naturally gives rise to the log-Laplace transform of a special random variable (Catoni, 2007). Importantly the log-Laplace transform has a temperature parameter $\lambda$ which is not constrained to be $\lambda = 1$. We investigate the relationship of this temperature parameter to cold posteriors.

In summary, our contributions are the following:

- Through detailed experiments for the Laplace approximation to the posterior, we show that PAC-Bayes bounds correlate with out-of-sample performance for different values of the temperature parameter $\lambda$. This might indicate that the temperature in the cold-posterior literature coincides with the temperature of the log-Laplace transform, and motivate Bayesian practitioners to use heuristics when targeting Frequentist metrics.

- Contrary to Wenzel et al. (2020), we find that the coldest temperature (such that the posterior is a Dirac delta centered on a MAP estimate of the weights) is empirically almost always optimal in terms of test accuracy. PAC-Bayes bounds track and predict this behaviour. However, the negative log-likelihood and the Expected Calibration Error (ECE) have a more complex behaviour. Contrary to prior work (Wenzel et al., 2020; Noci et al., 2021), this highlights that the evaluation metric choice plays an important role when discussing the cold-posterior effect. More importantly, we show that to improve the test ECE or NLL one typically needs to reduce the test accuracy.

- We derive a PAC-Bayes bound for the case of the widely used generalized Gauss–Newton Laplace approximations to the posterior. Contrary to prior work (Bachmann et al., 2022; Aitchison, 2021) our bound implies that $\lambda$ does not simply fix a misspecified prior or likelihood. For a fixed target test risk, likelihood and prior, the required $\lambda$ varies due to the stochasticity of the inference procedure and the loss landscape shape.

We also include a detailed FAQ section in the Appendix.

## 2 COLD POSTERIOR EFFECT: MISSPECIFIED AND NON-ASYMPTOTIC SETTING

We denote the learning sample $(X, Y) = \{(\boldsymbol{x}_i, y_i)\}_{i=1}^n \in (\mathcal{X} \times \mathcal{Y})^n$, that contains $n$ input-output pairs. Observations $(X, Y)$ are assumed to be sampled randomly from a distribution $\mathcal{D}$. Thus, we denote $(X, Y) \sim \mathcal{D}^n$ the i.i.d observation of $n$ elements. We consider loss functions $\ell : \mathcal{F} \times \mathcal{X} \times \mathcal{Y} \to \mathbb{R}$, where $\mathcal{F}$ is a set of predictors $f : \mathcal{X} \to \mathcal{Y}$. We also denote the risk $\mathcal{L}_{\mathcal{D}}^{\ell}(f) = \mathbf{E}_{(\boldsymbol{x},y) \sim \mathcal{D}} \ell(f, \boldsymbol{x}, y)$ and the empirical risk $\hat{\mathcal{L}}_{X,Y}^{\ell}(f) = (1/n) \sum_i \ell(f, \boldsymbol{x}_i, y_i)$. We consider two probability measures, the prior $\pi \in \mathcal{M}(\mathcal{F})$ and the approximate posterior $\hat{\rho} \in \mathcal{M}(\mathcal{F})$. Here, $\mathcal{M}(\mathcal{F})$ denotes the set of all probability measures on $\mathcal{F}$. We encounter cases where we make predictions using the approximate posterior predictive distribution $\mathbf{E}_{f \sim \hat{\rho}}[p(y|\boldsymbol{x}, f)]$. We will use two loss functions, the non-differentiable zero-one loss $\ell_{01}(f, \boldsymbol{x}, y) = \mathbb{I}(\arg\max_j f(\boldsymbol{x})_j \neq y)$, and the negative log-likelihood, which is a commonly used differentiable surrogate $\ell_{\text{nll}}(f, \boldsymbol{x}, y) = -\log(p(y|\boldsymbol{x}, f))$. We assume that outputs of $f$ form a probability distribution $p(y|\boldsymbol{x}, f)$ either through a Gaussian likelihood (in the case of regression) or using the softmax activation function (in the case of classification). Given the above, the Evidence Lower Bound (ELBO) has the following form

$$-\mathbf{E}_{f \sim \hat{\rho}} \hat{\mathcal{L}}_{X,Y}^{\ell_{\text{nll}}}(f) - \frac{1}{\lambda n} \text{KL}(\hat{\rho} \| \pi), \tag{1}$$

where $\lambda = 1$. Note that our temperature parameter $\lambda$ is the *inverse* of the one typically used in cold posterior papers. In this form $\lambda$ has a clearer interpretation as the temperature of a log-Laplace transform. Our setup is discussed in Wenzel et al. (2020), p3 Section 2.3, and used in Bachmann et al. (2022); Aitchison (2021). While Wenzel et al. (2020) use MCMC to conduct their experiments, we opt for the ELBO for analytical tractability. While Wenzel et al. (2020) temper by $\lambda$ both the likelihood and the prior, as discussed in Aitchison (2021) and Wenzel et al. (2020) the relevant setting for the ELBO is the one of (Eq. 1), where only the KL is tempered. One then typically models the posterior and prior distributions over weights using a parametric distribution (commonly a Gaussian) and optimizes the ELBO, using the reparametrization trick, to find the posterior distribution (Blundell et al., 2015; Khan et al., 2018). The cold posterior is the following observation:

> *Even though the ELBO has the form (1) with $\lambda = 1$, practitioners have found that much larger values $\lambda \gg 1$ typically result in better test time performance, for example a lower test misclassification rate and lower test negative log-likelihood.*

The starting point of our discussion will be thus to define the quantity that we care about in the context of Bayesian deep neural networks and cold posterior analyses. Concretely, in the setting of supervised prediction, what we often try to minimize is

$$\text{KL}(p_{\mathcal{D}}(y|\boldsymbol{x}) \| \mathbf{E}_{f \sim \hat{\rho}}[p(y|\boldsymbol{x}, f)]) = \mathbf{E}_{\boldsymbol{x}, y \sim \mathcal{D}} \left[ \ln \frac{p_{\mathcal{D}}(y|\boldsymbol{x})}{\mathbf{E}_{f \sim \hat{\rho}}[p(y|\boldsymbol{x}, f)]} \right], \tag{2}$$

the conditional relative entropy (Cover, 1999) between the true conditional distribution $p_{\mathcal{D}}(y|\boldsymbol{x})$ and the posterior predictive distribution $\mathbf{E}_{f \sim \hat{\rho}}[p(y|\boldsymbol{x}, f)]$. For example, this is implicitly the quantity that we minimize when optimizing classifiers using the cross-entropy loss (Masegosa, 2020; Morningstar et al., 2022). It is also on this and similar predictive metrics that the cold posterior appears. In the following we will outline the relationship between the ELBO, PAC-Bayes and (2).

### 2.1 ELBO

We assume a training sample $(X, Y) \sim \mathcal{D}^n$ as before, denote $p(\mathbf{w}|X, Y)$ the true posterior probability over predictors $f$ parameterized by $\mathbf{w}$ (typically weights for neural networks), and $\pi$ and $\hat{\rho}$ respectively the prior and variational posterior distributions as before. The ELBO results from the following calculations

$$
\begin{aligned}
\text{KL}(\hat{\rho}(\mathbf{w}) \| p(\mathbf{w}|X, Y)) &= \int \hat{\rho}(\mathbf{w}) \ln \frac{\hat{\rho}(\mathbf{w})}{p(\mathbf{w}|X, Y)} d\mathbf{w} = \int \hat{\rho}(\mathbf{w}) \ln \frac{\hat{\rho}(\mathbf{w}) p(Y|X)}{\pi(\mathbf{w}) p(Y|X, \mathbf{w})} d\mathbf{w} \\
&= \int \hat{\rho}(\mathbf{w}) \left[ -\ln p(Y|X, \mathbf{w}) + \ln \frac{\hat{\rho}(\mathbf{w})}{\pi(\mathbf{w})} + \ln p(Y|X) \right] d\mathbf{w} \\
&= -n \underbrace{\left( -\mathbf{E}_{f \sim \hat{\rho}} \hat{\mathcal{L}}_{X,Y}^{\ell_{\text{nll}}}(f) - \frac{1}{n} \text{KL}(\hat{\rho} \| \pi) \right)}_{\text{ELBO}} + \ln p(Y|X).
\end{aligned}
$$

Thus, maximizing the ELBO can be seen as minimizing the KL divergence between the true posterior and the variational posterior over the weights $\mathrm{KL}(\hat{\rho}(\mathbf{w})\|p(\mathbf{w}|X,Y))$. The true posterior distribution $p(\mathbf{w}|X,Y)$ gives more probability mass to predictors which are more likely given the training data, however these predictors do not necessarily minimize $\mathrm{KL}(p_{\mathcal{D}}(y|\boldsymbol{x})\|\mathbf{E}_{f\sim\hat{\rho}}[p(y|\boldsymbol{x},f)])$, the evaluation metric of choice (2) for supervised prediction. In the well-specified regime (where the true predictor $f^*$ is $f^* \in \mathcal{F}$) and when $n \to \infty$, the Blackwell–Dubins consistency theorem (Blackwell & Dubins, 1962) implies that the posterior quickly concentrates on the true set of parameters. In such cases, a more detailed analysis, such as a PAC-Bayesian one, is unnecessary as the posterior is akin to a Dirac delta mass at the true parameters. However neural networks do not operate in this regime. The existence of multiple minima hints that neural networks are misspecified, and the number of samples is small relative to the number of parameters.

Operating in the regime where $f^* \notin \mathcal{F}$ and where $n$ is (comparatively) small makes it important to derive a more precise certificate of generalization through a generalization bound, which directly bounds the true risk. In the following we focus on analyzing a PAC-Bayes bound in order to obtain insights into when the cold posterior effect occurs.

## 2.2 PAC-Bayes

We first look at the following bound denoted by $\mathcal{B}_{\mathrm{Alquier}}$. It was considered by Alquier et al. (2016) (Theorem 4.1); see also Theorem 1 by Masegosa (2020) for a statement under the same conditions.

**Theorem 1** ($\mathcal{B}_{\mathrm{Alquier}}$, Alquier et al., 2016). *Given a distribution $\mathcal{D}$ over $\mathcal{X} \times \mathcal{Y}$, a hypothesis set $\mathcal{F}$, a loss function $\ell : \mathcal{F} \times \mathcal{X} \times \mathcal{Y} \to \mathbb{R}$, a prior distribution $\pi$ over $\mathcal{F}$, real numbers $\delta \in (0,1]$ and $\lambda > 0$, with probability at least $1 - \delta$ over the choice $(X,Y) \sim \mathcal{D}^n$, we have for all $\hat{\rho}$ on $\mathcal{F}$*

$$\mathbf{E}_{f\sim\hat{\rho}}\mathcal{L}_{\mathcal{D}}^{\ell}(f) \le \mathbf{E}_{f\sim\hat{\rho}}\hat{\mathcal{L}}_{X,Y}^{\ell}(f) + \frac{1}{\lambda n}\left[\mathrm{KL}(\hat{\rho}\|\pi) + \ln\frac{1}{\delta} + \Psi_{\ell,\pi,\mathcal{D}}(\lambda,n)\right]$$

$$where\ \Psi_{\ell,\pi,\mathcal{D}}(\lambda,n) = \ln\mathbf{E}_{f\sim\pi}\mathbf{E}_{X',Y'\sim\mathcal{D}^n}\exp\left[\lambda n\left(\mathcal{L}_{\mathcal{D}}^{\ell}(f) - \hat{\mathcal{L}}_{X',Y'}^{\ell}(f)\right)\right].$$

There are three different terms in the above bound. The empirical risk term $\mathbf{E}_{f\sim\hat{\rho}}\hat{\mathcal{L}}_{X,Y}^{\ell}(f)$ is the empirical mean of the loss of the classifier over all training samples. The KL term $1/(\lambda n)\mathrm{KL}(\hat{\rho}\|\pi)$ is the complexity of the model, which in this case is measured as the KL-divergence between the posterior and prior distributions. The Moment term $1/(\lambda n)\Psi_{\ell,\pi,\mathcal{D}}(\lambda,n)$ is the log-Laplace transform of the difference between the risk and empirical risk for a reversal of the temperature. We will keep the name "Moment" in the following. Note that unless stated otherwise (for example in Section 4), we can make some assumption on the risk to ensure that the Moment term is bounded. Such common assumptions include that the risk is a sub-Gaussian or sub-Gamma random variable under the prior $\pi$ and the distribution $\mathcal{D}$ (see eg Germain et al., 2016) or more generally sub-Weibull (Vladimirova et al., 2019; 2020). Using a PAC-Bayes bound together with Jensen's inequality, one can bound (2) directly as follows

$$\mathrm{KL}(p_{\mathcal{D}}(y|\boldsymbol{x})\|\mathbf{E}_{f\sim\hat{\rho}}[p(y|\boldsymbol{x},f)]) = \mathbf{E}_{\boldsymbol{x},y\sim\mathcal{D}}\left[\ln\frac{p_{\mathcal{D}}(y|\boldsymbol{x})}{\mathbf{E}_{f\sim\hat{\rho}}[p(y|\boldsymbol{x},f)]}\right]$$

$$= \mathbf{E}_{\boldsymbol{x},y\sim\mathcal{D}}[-\ln\mathbf{E}_{f\sim\hat{\rho}}[p(y|\boldsymbol{x},f)]] + \mathbf{E}_{\boldsymbol{x},y\sim\mathcal{D}}[\ln p_{\mathcal{D}}(y|\boldsymbol{x})]$$

$$\le \mathbf{E}_{\boldsymbol{x},y\sim\mathcal{D}}[\mathbf{E}_{f\sim\hat{\rho}}[-\ln p(y|\boldsymbol{x},f)]] + \mathbf{E}_{\boldsymbol{x},y\sim\mathcal{D}}[\ln p_{\mathcal{D}}(y|\boldsymbol{x})]$$

$$\le \underbrace{\mathbf{E}_{f\sim\hat{\rho}}\hat{\mathcal{L}}_{X,Y}^{\ell_{\mathrm{nll}}}(f) + \frac{1}{\lambda n}\left[\mathrm{KL}(\hat{\rho}\|\pi) + \ln\frac{1}{\delta} + \Psi_{\ell_{\mathrm{nll}},\pi,\mathcal{D}}(\lambda,n)\right]}_{\text{PAC-Bayes}} + \mathbf{E}_{\boldsymbol{x},y\sim\mathcal{D}}[\ln p_{\mathcal{D}}(y|\boldsymbol{x})].$$

The last line holds under the conditions of Theorem 1 and in particular with probability at least $1-\delta$ over the choice $(X,Y) \sim \mathcal{D}^n$. Notice here the presence of the temperature parameter $\lambda \ge 0$, which need not be $\lambda = 1$.

> *In particular it is easy to see that maximizing the ELBO is equivalent to minimizing a PAC-Bayes bound for $\lambda = 1$, which might not necessarily be optimal for a finite sample size. More specifically even for exact inference, where $\mathbf{E}_{\mathbf{w}\sim\hat{\rho}}[p(y|\boldsymbol{x},\mathbf{w})]|_{\hat{\rho}=p(\mathbf{w}|X,Y)} = p(y|\boldsymbol{x},X,Y)$, the Bayesian posterior predictive distribution does not necessarily minimize $\mathrm{KL}(p_{\mathcal{D}}(y|\boldsymbol{x})\|\mathbf{E}_{f\sim\hat{\rho}}[p(y|\boldsymbol{x},f)])$.*

### 2.3 CLASSIFICATION TASKS

For classification tasks, we are typically mainly interested in achieving low expected zero-one risk $\mathbf{E}_{f\sim\hat{\rho}}\mathcal{L}_{\mathcal{D}}^{\ell_{01}}(f)$. The ELBO objective is not directly related to this risk. However in the PAC-Bayesian literature there exist bounds specifically adapted to it. In the following we will use one of the tightest and most commonly used bounds, the "Catoni" bound, denoted $\mathcal{B}_{\text{Catoni}}$ from Catoni (2007) Theorem 1.2.6.

**Theorem 2** ($\mathcal{B}_{\text{Catoni}}$, Catoni, 2007). *Given a distribution $\mathcal{D}$ over $\mathcal{X}\times\mathcal{Y}$, a hypothesis set $\mathcal{F}$, the 0-1 loss function $\ell_{01}:\mathcal{F}\times\mathcal{X}\times\mathcal{Y}\to[0,1]$, a prior distribution $\pi$ over $\mathcal{F}$, a real number $\delta\in(0,1]$, and a real number $\lambda>0$, with probability at least $1-\delta$ over the choice of $(X,Y)\sim\mathcal{D}^n$, we have*

$$\forall\hat{\rho}\text{ on }\mathcal{F}:\mathbf{E}_{f\sim\hat{\rho}}\mathcal{L}_{\mathcal{D}}^{\ell_{01}}(f)\leq\Phi_\lambda^{-1}\left(\mathbf{E}_{f\sim\hat{\rho}}\hat{\mathcal{L}}_{X,Y}^{\ell_{01}}(f)+\frac{1}{\lambda n}\left[\text{KL}(\hat{\rho}||\pi)+\ln\frac{1}{\delta}\right]\right), \quad (3)$$

*where $\Phi_\lambda^{-1}(x)=\frac{1-e^{-\lambda x}}{1-e^{-\lambda}}$.*

Similarly to the Alquier bound, the empirical risk term is the empirical mean of the loss of the classifier over all training samples. The KL term is the complexity of the model, which in this case is measured as the KL-divergence between the posterior and prior distributions. The Moment term has been absorbed in this case in the function $\Phi_\lambda^{-1}(x)=\frac{1-e^{-\lambda x}}{1-e^{-\lambda}}$.

### 2.4 SAFE-BAYES AND OTHER RELEVANT WORK

After identifying two sources of misspecification in standard Bayesian inference, Grünwald & Langford (2007) proposed a solution, through an approach which they named Safe-Bayes (Grünwald, 2012; Grünwald & Van Ommen, 2017). Safe-Bayes corresponds to finding a temperature parameter $\lambda$ for a generalized (tempered) posterior distribution with $\lambda$ possibly different than 1. The optimal value of $\lambda$ is found by taking a sequential view of Bayesian inference, and for a Cèsaro averaged posterior, which is an average of the posteriors at different optimization steps, and which doesn't coincide with the standard posterior. The analysis of Grünwald (2012); Grünwald & Van Ommen (2017) is also restricted to the case where $\lambda<1$. By contrast we provide an analytical expression of the bound on true risk, given $\lambda$, and also numerically investigate the case of $\lambda>1$. Our analysis thus provides intuition regarding which parameters (for example the curvature) might result in cold posteriors. Catoni (2007) discusses the optimal value of the temperature $\lambda$ for PAC-Bayes bounds, for *fixed* priors and posteriors. By contrast we investigate the case where the posterior is optimized for different $\lambda$ and which is the relevant one for the cold-posterior literature. Germain et al. (2016) find that minimizing a PAC-Bayesian generalization risk bound maximizes the Bayesian marginal likelihood. However they only investigate the case where $\lambda=1$.

## 3 EXPERIMENTS

### 3.1 EXPERIMENTAL SETUP

The ELBO (1) is minimized at the probability density $\rho^\star(f)$ given by: $\rho^\star(f):=\pi(f)e^{-\lambda n\hat{\mathcal{L}}_{X,Y}^{\ell_{\text{nll}}}(f)}/\mathbf{E}_{f\sim\pi}\left[e^{-\lambda n\hat{\mathcal{L}}_{X,Y}^{\ell_{\text{nll}}}(f)}\right]$ (Catoni, 2007). We will use the Laplace approximation to the posterior in our experiments. This is equivalent to approximating $\lambda n\hat{\mathcal{L}}_{X,Y}^{\ell_{\text{nll}}}(f)$ using a second order Taylor expansion around a minimum $\mathbf{w}_{\hat{\rho}}$, such that $\lambda n\hat{\mathcal{L}}_{X,Y}^{\ell_{\text{nll}}}(f_\mathbf{w})\approx\lambda n\hat{\mathcal{L}}_{X,Y}^{\ell_{\text{nll}}}(f_{\mathbf{w}_{\hat{\rho}}})+\lambda n(\mathbf{w}-\mathbf{w}_{\hat{\rho}})^\top\frac{1}{2}\nabla\nabla\hat{\mathcal{L}}_{X,Y}^{\ell_{\text{nll}}}(f_\mathbf{w})|_{\mathbf{w}=\mathbf{w}_{\hat{\rho}}}(\mathbf{w}-\mathbf{w}_{\hat{\rho}})$. Assuming a Gaussian prior $\pi=\mathcal{N}(0,\sigma_\pi^2\mathbf{I})$, the Laplace approximation to the posterior $\hat{\rho}$ is again a Gaussian

$$\hat{\rho}=\mathcal{N}\left(\mathbf{w}_{\hat{\rho}},\left(\lambda\mathbf{H}+\frac{1}{\sigma_\pi^2}\mathbf{I}\right)^{-1}\right)$$

where $\mathbf{H}$ is the network Hessian $\mathbf{H}=n\nabla\nabla\hat{\mathcal{L}}_{X,Y}^{\ell_{\text{nll}}}(f_\mathbf{w})|_{\mathbf{w}=\mathbf{w}_{\hat{\rho}}}$. This Hessian is generally infeasible to compute in practice for modern deep neural networks, such that many approaches employ the generalized Gauss–Newton (GGN) approximation $\mathbf{H}^{\text{GGN}}:=\sum_{i=1}^n\mathcal{J}_\mathbf{w}(\boldsymbol{x}_i)^\top\boldsymbol{\Lambda}(\boldsymbol{y}_i;f_i)\mathcal{J}_\mathbf{w}(\boldsymbol{x}_i)$,

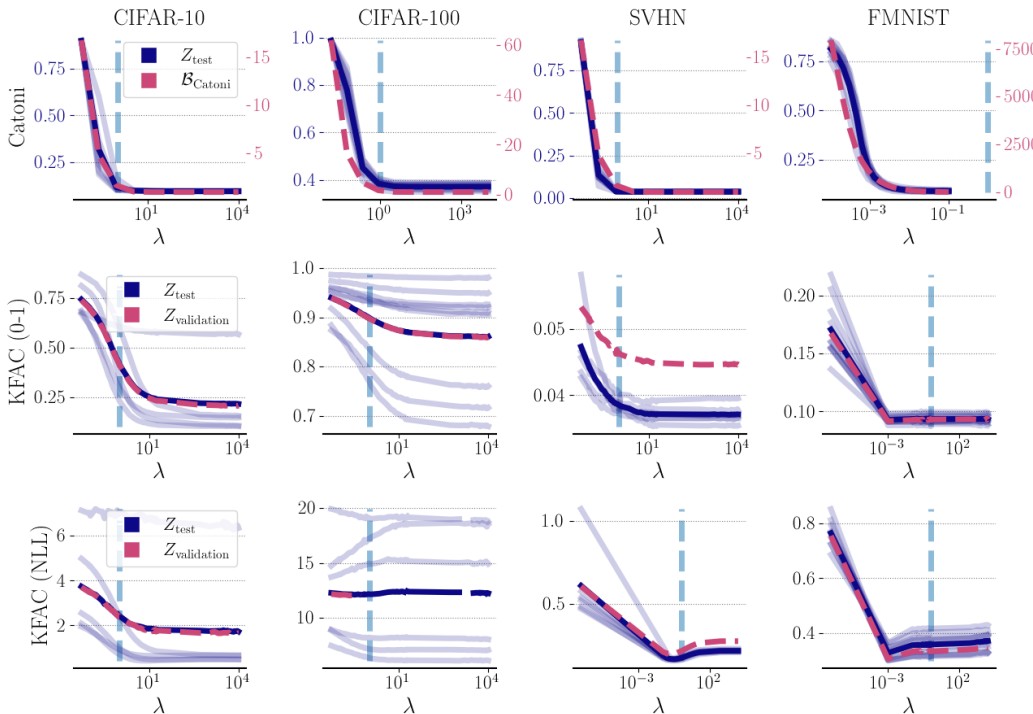

Figure 2: Test 0-1 Loss ▬▬▬ mean, as well as 10 MAP trials ▬▬▬, along with the mean generalization certificate ▬ ▬ ▬ (we denote $\lambda = 1$ by ▬ ▬ ▬): $\mathcal{B}_{\text{Catoni}}$ PAC-Bayes bound 0-1 loss (top), standard KFAC Laplace 0-1 loss (middle) and standard KFAC NLL (bottom). The $\mathcal{B}_{\text{Catoni}}$ PAC-Bayes bound closely tracks the test 0-1 loss. For the standard KFAC posteriors the test and validation 0-1 loss behave similar to the Catoni case, with a rapid improvement as $\lambda \uparrow$ followed by a plateau. Coldest posteriors $\lambda \gg 1$ are almost always best.

where $\mathcal{J}_{\mathbf{w}}(\boldsymbol{x})$ is the network per-sample Jacobian $[\mathcal{J}_{\mathbf{w}}(\boldsymbol{x})]_c = \nabla_{\mathbf{w}} f_c(\boldsymbol{x}; \mathbf{w}_{\hat{\rho}})$, and $\boldsymbol{\Lambda}(\boldsymbol{y}; f) = -\nabla_{ff}^2 \log p(\boldsymbol{y}; f)$ is the per-input noise matrix (Kunstner et al., 2019). We will use two simplified versions of the GGN

- An isotropic approximation with variance $\sigma_{\hat{\rho}}^2(\lambda)$ such that $\frac{1}{\sigma_{\hat{\rho}}^2(\lambda)} = \frac{\lambda h}{d} + \frac{1}{\sigma_\pi^2}$, where $h = \sum_{i,j,k} g(i,k)(\nabla_{\mathbf{w}} f_k(\boldsymbol{x}_i; \mathbf{w}_{\hat{\rho}})_j)^2$ is the trace of the Gauss–Newton approximation to the Hessian, with $g(i,k) = [\boldsymbol{\Lambda}(\boldsymbol{y}_i; f)]_{kk}$.
- The Kronecker-Factorized Approximate Curvature (KFAC) (Martens & Grosse, 2015) approximation, which retains only a block diagonal part of the GGN.

When making predictions, we use the posterior predictive distribution $\mathbf{E}_{\mathbf{w} \sim \hat{\rho}}[p(y|\boldsymbol{x}, f_{\mathbf{w}})]$ of the *full neural network model*, meaning that samples from $\hat{\rho}$ are inputted to the full neural network. Since the 0-1 loss is not differentiable, the posterior estimated with the cross entropy loss will be used for classification problems.

We have tested extensively in realistic classification tasks. We used the CIFAR-10, CIFAR-100 (Krizhevsky & Hinton, 2009), SVHN (Netzer et al., 2011) and FashionMnist (Xiao et al., 2017) datasets. In all experiments, we split the dataset into three sets. These three are the typical prediction tasks sets: training set $Z_{\text{train}}$, testing set $Z_{\text{test}}$, and validation set $Z_{\text{validation}}$. We use Monte Carlo sampling to estimate the Empirical Risk term ($f \sim \hat{\rho}$). For the isotropic Laplace approximation, and a Gaussian isotropic prior, the KL divergence has a simple analytical expression $\text{KL}(\hat{\rho}||\pi) = \frac{1}{2}\left(d\frac{\sigma_{\hat{\rho}}^2(\lambda)}{\sigma_\pi^2} + \frac{1}{\sigma_\pi^2}\|\mathbf{w}_{\hat{\rho}} - \mathbf{w}_\pi\|^2 - d - d\ln\sigma_{\hat{\rho}}^2(\lambda) + d\ln\sigma_\pi^2\right)$. PAC-Bayes bounds require correct control of the prior mean as the $\ell_2$ distance between prior and posterior means in the KL term is often the dominant term in the bound. To control this distance, we follow a variation of

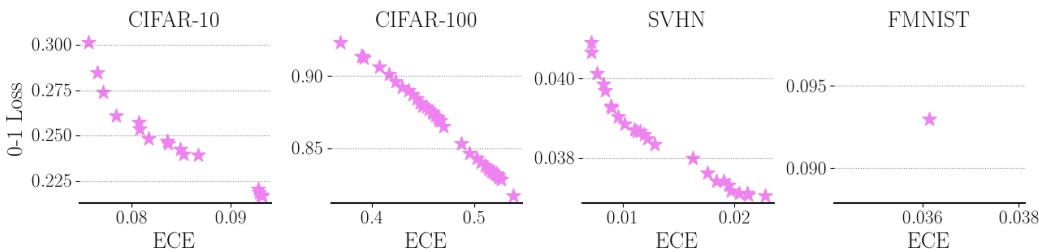

Figure 3: The Pareto front of the test ECE compared to test 0-1 loss. Apart from the case of FMNIST we see a clear trade-off between 0-1 loss and ECE. As such there seems to be no $\lambda$ that simultaneously doesn't hurt the test 0-1 loss while improving the ECE, which is the most pertinent task, in light of the different behaviours of the different metrics with respect to $\lambda$.

the approach in Dziugaite et al. (2021) to constructing our classifiers. We first use $Z_{\text{train}}$ to find a prior mean $\mathbf{w}_\pi$. We then set the posterior mean equal to the prior mean $\mathbf{w}_{\hat\rho} = \mathbf{w}_\pi$ but evaluate the r.h.s of the bounds on $Z_{\text{validation}}$. Note that in this way $\|\mathbf{w}_{\hat\rho} - \mathbf{w}_\pi\|_2^2 = 0$, while the bound is still valid since the prior is independent from the evaluation set $X, Y = Z_{\text{validation}}$. For the CIFAR-10, CIFAR-100, and SVHN datasets, we use a WideResNet22 (Zagoruyko & Komodakis, 2016), with Fixup initialization (Zhang et al., 2019). For the FashionMnist dataset, we use a convolutional architecture with three convolutional layers, followed by two fully connected non-linear layers. More details on the experimental setup can be found in the Appendix.

## 3.2 CLASSIFICATION EXPERIMENTS

We find ten MAP estimates for the neural network weights of the CIFAR-10, CIFAR-100, SVHN and FMNIST datasets by training on $Z_{\text{train}}$ using SGD. We then fit an Isotropic Laplace approximation to each MAP estimate using $X, Y = Z_{\text{validation}}$. For different values of $\lambda$ we then estimate the Catoni bound (Theorem 2) using $Z_{\text{validation}}$. We also estimate the *test* 0-1 Loss, negative log-likelihood (NLL) and the Expected Calibration Error (ECE) (Naeini et al., 2015) of the posterior predictive on $Z_{\text{test}}$. We use the prior variance $\sigma_\pi^2 = 0.1$, as optimizing the marginal likelihood leads to $\sigma_\pi^2 \approx 0$ which is not relevant for BNNs. We also test a standard KFAC Laplace setup. Specifically we fit the KFAC Laplace on $Z_{\text{train}}$ and also choose the prior through the marginal likelihood. In this case, we estimate a standard validation set bound using the validation set $Z_{\text{validation}}$ (instead of a PAC-Bayes bound) as from the literature we know that any PAC-Bayes bound will be vacuous (larger than 1) as we do not control $\|\mathbf{w}_{\hat\rho} - \mathbf{w}_\pi\|_2^2$. We plot the results for all datasets in Figure 2. The Catoni bound correlates tightly with test 0-1 Loss for all datasets and we plot this correlation in Figure 1(a). Contrary to Wenzel et al. (2020) in terms of test 0-1 Loss, the MAP estimate (obtained where $\lambda \gg 1$ and the posterior is "coldest") is almost always optimal. This behaviour is replicated in the "KFAC" case. This result is more coherent than the one in Wenzel et al. (2020) (Figure 1) where the coldest temperature 0-1 test loss and the MAP and the 0-1 test loss don't match. It highlights that in a tightly controlled setting, often Bayesian approaches don't improve at all over deterministic ones in terms of test 0-1 loss, *for any temperature $\lambda$*.

We plot in Figure 2 (bottom row) the NLL for the KFAC case. Even without data augmentation and even when we optimize the prior variance using the marginal likelihood, we find that all three cases of temperatures (cold posterior, warm posterior, as well as posterior with $\lambda = 1$) can be optimal, for varying datasets. *This highlights the importance of the choice of the evaluation metric when discussing the cold posterior effect, as results can vary significantly depending on our choice.* Specifically one can see in Kapoor et al. (2022) p18 and Aitchison (2021) p9 that different metrics have different behaviours and/or minima with respect to $\lambda$.

We then ask the more pertinent question: "If the Laplace approximation doesn't in general improve the test 0-1 loss, can it (for some value of $\lambda$) retain the same test 0-1 loss while improving calibration?". In Figure 3 we see that in our experiments in most cases we couldn't find such a temperature $\lambda$. Apart from FMNIST there seems to be a clear tradeoff between test 0-1 loss and calibration error.

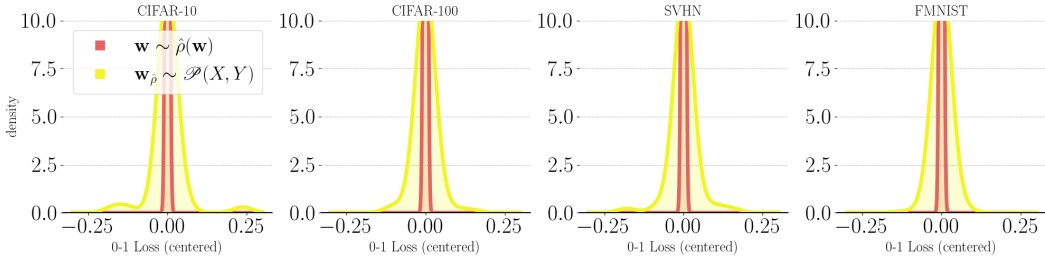

Figure 4: Intragroup variability of the test 0-1 loss for all Laplace approximations ■ $\mathbf{w}_{\hat{\rho}} \sim \mathscr{P}(X, Y)$, as well as the test 0-1 loss variability for each Laplace approximation due to the MC approximation to the Laplace predictive■ $\mathbf{w} \sim \hat{\rho}(\mathbf{w})$. We display the combined variability for all values of $\lambda \in [10^{-7}, 10^4]$. The intragroup test 0-1 variability cannot be explained by the variability of each Laplace predictive due to the MC approximation. This large intragroup variability contradicts the assumption that $\lambda$ simply adjusts for the aleatoric uncertainty of the data. Intuitively, different MAP estimates result in different predictive functions, which yield different test 0-1 losses for a fixed prior, $\lambda$, and likelihood. Alternatively for a fixed prior, likelihood, and *target test 0-1 loss* the required value of $\lambda$ depends on the MAP estimate. As MAP estimates are usually found using a stochastic algorithm $\mathscr{P}(X, Y)$ finding a $\lambda$ that consistently achieves the target test 0-1 loss might be difficult.

## 4 THE TEMPERATURE $\lambda$, ALEATORIC UNCERTAINTY AND CURVATURE

In light of our empirical results, it would be interesting to derive an analytical form that elucidates the important variables that affect the bound. However, PAC-Bayes objectives are difficult to analyze theoretically for the non-convex case. Thus in the following we make a number of simplifying assumptions. The Laplace approximation with the Generalized Gauss-Newton approximation to the Hessian corresponds to a linearization of the neural network around the MAP estimate $\mathbf{w}_{\hat{\rho}} \in \mathbb{R}^d$ (Immer et al., 2021) $f_{\text{lin}}(\boldsymbol{x}; \mathbf{w}) = f(\boldsymbol{x}; \mathbf{w}_{\hat{\rho}}) + \nabla_{\mathbf{w}} f(\boldsymbol{x}; \mathbf{w}_{\hat{\rho}})^{\top}(\mathbf{w} - \mathbf{w}_{\hat{\rho}})$.

When analyzing minima of the loss landscape linearization is reasonable even without assuming infinite width Zancato et al. (2020); Maddox et al. (2021). For appropriate modelling choices, we aim at deriving a bound for this linearized model. We adopt the aforementioned linear form together with the Gaussian likelihood, yielding $\ell_{\text{nll}}(\mathbf{w}, \boldsymbol{x}, y) = \frac{1}{2} \ln(2\pi\sigma^2) + \frac{1}{2\sigma^2}(y - f(\boldsymbol{x}; \mathbf{w}_{\hat{\rho}}) - \nabla_{\mathbf{w}} f(\boldsymbol{x}; \mathbf{w}_{\hat{\rho}})^{\top}(\mathbf{w} - \mathbf{w}_{\hat{\rho}}))^2$. We also make the following modeling choices: 1) Prior over weights: $\mathbf{w} \sim \mathcal{N}(\mathbf{w}_{\pi}, \sigma_{\pi}^2 \mathbf{I})$. 2) Gradients as Gaussian mixture: $\nabla_{\mathbf{w}} f(\boldsymbol{x}; \mathbf{w}_{\hat{\rho}}) \sim \sum_{i=1}^{k} \phi_i \mathcal{N}(\boldsymbol{\mu}_i, \sigma_{\boldsymbol{x}i}^2 \mathbf{I})$. 3) Labeling function: $y = f(\boldsymbol{x}; \mathbf{w}_{\hat{\rho}}) + \nabla_{\mathbf{w}} f(\boldsymbol{x}; \mathbf{w}_{\hat{\rho}})^{\top}(\mathbf{w}_* - \mathbf{w}_{\hat{\rho}}) + \epsilon$, where $\epsilon \sim \mathcal{N}(0, \sigma_{\epsilon}^2)$. We also assume that we have a deterministic estimate of the posterior weights $\mathbf{w}_{\hat{\rho}}$ *which we keep fixed*, and we model the posterior as $\hat{\rho} = \mathcal{N}(\mathbf{w}_{\hat{\rho}}, \sigma_{\hat{\rho}}^2(\lambda)\mathbf{I})$, similarly to our experimental section. Therefore estimating the posterior corresponds to estimating the variance $\sigma_{\hat{\rho}}^2(\lambda)$.

**Proposition 1** ($\mathcal{B}_{\text{approximate}}$). *With the above modeling choices, and given a distribution $\mathcal{D}$ over $\mathcal{X} \times \mathcal{Y}$, real numbers $\delta \in (0, 1]$ and $\lambda \in (0, \frac{1}{c})$ with $c = 2n\sigma_{\boldsymbol{x}}^2\sigma_{\pi}^2$, with probability at least $1 - \delta$ over the choice $(X, Y) \sim \mathcal{D}^n$, we have*

$$\mathbf{E}_{\mathbf{w}\sim\hat{\rho}}\mathcal{L}_{\mathcal{D}}^{\ell_{\text{nll}}}(\mathbf{w})$$

$$\leq \underbrace{\frac{\|\boldsymbol{y} - f(\mathbf{X}; \mathbf{w}_{\hat{\rho}})\|_2^2}{2n\,\sigma^2} + \left(\frac{\lambda\,h}{d\,\sigma^2} + \frac{1}{\sigma_{\pi}^2}\right)^{-1} \frac{h}{2n\,\sigma^2} + \frac{1}{2}\ln(2\pi\,\sigma^2)}_{\text{Empirical Risk}} + \underbrace{\frac{\sigma_{\boldsymbol{x}}^2(\sigma_{\pi}^2 d + \|\mathbf{w}_*\|_2^2)}{1 - 2\lambda\,n\sigma_{\boldsymbol{x}}^2\sigma_{\pi}^2} + \sigma_{\epsilon}^2}_{\text{Moment}} +$$

$$\underbrace{\frac{1}{\lambda\,n}\left[\frac{1}{2}\left(\frac{d}{\sigma_{\pi}^2}\frac{1}{\frac{\lambda\,h}{d\,\sigma^2} + \frac{1}{\sigma_{\pi}^2}} + \frac{1}{\sigma_{\pi}^2}\|\mathbf{w}_{\hat{\rho}} - \mathbf{w}_{\pi}\|_2^2 - d - d\ln\frac{1}{\frac{\lambda\,h}{d\,\sigma^2} + \frac{1}{\sigma_{\pi}^2}} + d\ln\sigma_{\pi}^2\right) + \ln\frac{1}{\delta}\right]}_{\text{KL}}$$

*where $h = \sum_i \sum_j (\nabla_{\mathbf{w}} f(\boldsymbol{x}_i; \mathbf{w}_{\hat{\rho}})_j)^2$ is the curvature parameter, $\sigma_{\boldsymbol{x}}^2 = \sum_{j=1}^{k} \phi_j \sigma_{\boldsymbol{x}j}^2$ is the posterior gradient variance, and $\sigma^2$ is the variance of the likelihood function.*

We now make a number of observations regarding Proposition 1. Here, $h$ is the trace of the Hessian under the Gauss–Newton approximation (without a scaling factor $n$). Under the PAC-Bayesian modeling of the risk, cold posteriors are the result of a complex interaction between various parameters resulting from 1) our *model* such as the prior variance $\sigma_{\pi}^2$ and prior mean $\mathbf{w}_{\pi}$, and 2) our *data* $\sigma_{\boldsymbol{x}}^2$ and $\mathbf{w}_*$ (the curvature of the minimum $h$ and the MAP estimate $\mathbf{w}_{\hat{\rho}}$ depend on the deep neural network architecture, the optimization procedure and the data). Contrary to prior work (Bachmann et al., 2022; Nabarro et al., 2022; Aitchison, 2021) our bound suggests that $\lambda$ cannot be seen as simply fixing a mispecified likelihood variance $\sigma^2$ or prior variance $\sigma_{\pi}^2$. In particular it does not simply rescale the aforementioned quantities. Furthermore, our bound implies that even for a fixed prior and likelihood the same $\lambda$ can imply different test risk based on the properties of each MAP estimate such as the curvature $h$ and the distance from initialization $||\mathbf{w}_{\hat{\rho}} - \mathbf{w}_{\pi}||_2^2$.

We can observe this in our empirical data. We fit a Laplace approximation with a similar procedure as the KFAC case of Figure 2 but using the Isotropic approximation to the posterior, for the CIFAR-10, CIFAR-100, SVHN and FMNIST datasets. We keep the prior variance fixed $\sigma_{\pi}^2 = 0.1$. For each value of $\lambda \in [10^{-7}, 10^4]$ we compute the test 0-1 loss of all the Laplace approximations at the different MAP estimates $\mathbf{E}_{f \sim \hat{\rho}(i)} \hat{\mathcal{L}}_{X_{\text{test}}, Y_{\text{test}}}^{\ell_{01}}(f), i \in [1, ..., M]$ where $M$ is the number of MAP estimates, and then the mean of these test 0-1 losses $(1/M) \sum_{i=1}^{M} \mathbf{E}_{f \sim \hat{\rho}(i)} \hat{\mathcal{L}}_{X_{\text{test}}, Y_{\text{test}}}^{\ell_{01}}(f)$. We then compute the residuals of the test 0-1 loss of each Laplace approximation with respect to the aforementioned mean. This gives us a measure of how much the test 0-1 loss of each Laplace approximation deviates from the mean test 0-1 loss of all Laplace approximations. We plot the inferred distribution of residuals for all $\lambda$ combined in Figure 4. For the Laplace approximation at each MAP we are using 100 Monte Carlo samples to approximate the predictive. As such, it is important to plot as a sanity check the variability we would expect simply from the MC approximation of each predictive for each $\mathbf{E}_{f \sim \hat{\rho}(i)} \hat{\mathcal{L}}_{X_{\text{test}}, Y_{\text{test}}}^{\ell_{01}}(f)$. Specifically we model this as a Gaussian distribution with $\sigma = 0.005$ such that $\mathbb{P}(|\mathbf{E}_{f \sim \hat{\rho}} \hat{\mathcal{L}}_{X_{\text{test}}, Y_{\text{test}}}^{\ell_{01}}(f) - (1/N_{\text{MC}}) \sum_{i=1}^{N_{\text{MC}}} \hat{\mathcal{L}}_{X_{\text{test}}, Y_{\text{test}}}^{\ell_{01}}(f_i)| \leq 0.01) \geq 0.95$ where $f_i \sim \hat{\rho}$ (*though we stress that this modelling comes from past experience and we didn't have time to precisely estimate this error*). In Figure 4 we see significant deviations from $(1/M) \sum_{i=1}^{M} \mathbf{E}_{f \sim \hat{\rho}(i)} \hat{\mathcal{L}}_{X_{\text{test}}, Y_{\text{test}}}^{\ell_{01}}(f)$ which can't be explained from the MC approximation of each predictive. Most of the variability comes from moderate values of $\lambda$ as for large values the Laplace approximations degenerate to Dirac masses on the MAP estimates, and all our MAP estimates have approximately the same test 0-1 loss. Alternatively for small values of $\lambda$ all Laplace approximations have 90% test 0-1 loss.

Assume that different MAP estimates $\mathbf{w}_{\hat{\rho}}$ are generated from some randomized algorithm $\mathbf{w}_{\hat{\rho}} \sim \mathscr{P}(X, Y)$ where $X, Y$ are the training data. Our results imply that *even for the same dataset*, a fixed prior and likelihood, it might be difficult to find a single value of $\lambda$ that gives *a target test loss*, simply because of the stochasticity of $\mathbf{w}_{\hat{\rho}} \sim \mathscr{P}(X, Y)$.

## 5 DISCUSSION

We argued that Bayesian inference does not readily provide high probability guarantees on out-of-sample performance, leading to inconsistencies such as the cold-posterior effect. We hope that this will motivate Bayesian practitioners to use heuristics "guilt-free" when targeting Frequentist performance metrics, or target contraction to the true posterior. Furthermore, our empirical results on the ECE point towards the need for the use of Pareto curves when evaluating Bayesian approaches. Finally, it would be interesting to see how our results from Section 4 translate to the MCMC setting.

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
