# OpenReview forum: "Cold Posteriors through PAC-Bayes"
_ICLR.cc/2023/Conference — Submitted to ICLR 2023_

### Official Review · Reviewer_jFkt · 2022-10-21

**Confidence:** 3
**Correctness:** 4
**Technical Novelty And Significance:** 2
**Empirical Novelty And Significance:** 2
**Recommendation:** 5

**Clarity, Quality, Novelty And Reproducibility:**

        The paper is well written and in general clear.


**Strength And Weaknesses:**

Strength:

        - Nice theoretical results and experimental evaluation.

Weaknesses:

        - The results obtained may explain the wide variety of explanations observed in the literature for the cold posterior effect, but the paper does not provide insights about to address this problem or improve current models. This is a bit disappointing in this regard.



**Summary Of The Paper:**

        This paper analyzes the cold posterior effect in Bayesian neural networks. This effect is simply that these models perform better at test time if the posterior is sharpened artificially. The authors argue that there are several works form the literature analyzing the causes of this, with mixed results. In this paper the authors use generalization PAC-Bayes bounds to study the cold posterior effect. They show that these bounds correlate well with the generalization error in terms of the parameter lambda, controlling the amount of sharpness. Then, they analyze the influence of these bounds on such parameter, in the case of linearized models. The results show that the bounds incorporates complex interactions, which may explain the mixed results observed in the literature.


**Summary Of The Review:**

  I believe that this is a nice paper with nice theoretical results and good empirical observations. However, I am a bit disappointed in the sense that it does not provide any improvement over the current methods that are provided. This questions the practical utility of the theoretical results obtained and the observations made.

---

> ### Author Response · Authors · 2022-11-06
> **Reply jFkt**
>
> Thank you for your review!
>
> We propose that the entire premise of [1], comparing purely Bayesian approaches on Frequentist metrics is ill-founded.
>
> [1] Has been used as a poster-child for supposed "inherent problems" with Bayesian deep learning. Therefore, considerable effort has been put into refuting [1]. We ask the question: "Is it even correct to try to compete at Frequentist metrics with purely Bayesian approaches (such as the ELBO) when these have been traditionally used to infer the correct model posterior?" We argue that it might not be correct, which might save time for Bayesian deep learning practitioners. It might motivate them to use heuristics such as tempering "guilt free" or alternatively evaluate on different metrics such as posterior contraction which are better founded for purely Bayesian approaches.
>
> For the case of the K-FAC Laplace approximation we have furthermore shown empirically that being even a bit Bayesian does not help with any of the most popular Frequentist evaluation metrics (at least without hurting another). This is a stronger result than the original cold posterior paper [1], which supposedly shows small improvements for some stochasticity. Even common heuristics don't seem to work here, which poses the important question of how can we make the Laplace approximation consistently improve upon the MAP. Furthermore, our PAC-Bayesian interpretation correctly predicts that the deterministic predictor is optimal.
>
> We kindly ask that the reviewer reevaluate his score as, even though the subject is tedious, we believe that it is important for the progress of Bayesian DL.
>
> [1]: Wenzel, Florian, et al. "How good is the bayes posterior in deep neural networks really?." arXiv preprint arXiv:2002.02405 (2020).

---

> > ### Comment · Reviewer_jFkt · 2022-11-17
> > **Response to reviwers**
> >
> > I would like to thank the reviewers for their response. However, I have decided to keep my score as such.

---

### Official Review · Reviewer_qVvM · 2022-10-23

**Confidence:** 4
**Correctness:** 2
**Technical Novelty And Significance:** 2
**Empirical Novelty And Significance:** 2
**Recommendation:** 3

**Clarity, Quality, Novelty And Reproducibility:**

**Clarity.** There were a few places in which I had trouble with the notation / development.
1. "Observations $(X,Y)$ are assumed..." Do you mean $(x_i, y_i)$ here?
2. I don't think it was ever explicitly said where the distribution $p(y \mid x, f)$ comes from (two sentences later normalizing model outputs is mentioned, but this should be made explicit).
3. There's some switching between considering distributions over functions and distributions over weights of neural networks. It would be good to settle on one.
4. Before section 2.1, $\hat\rho$ was "the posterior". But it seems like in Section 2.1 it is an approximation to the posterior. Can the authors clarify or standardize this?
5. In the second equality of the equations displayed at the bottom of page 3, the authors write:
$$\hat\rho(w) \log \frac{\hat\rho(w)}{p(w \mid X, Y)} = \hat\rho(w) \log \frac{\hat\rho(w) p(Y \mid X)}{\pi(w)p(Y \mid X, w)}.$$
It seems like $p(w \mid X, Y)$ has been expanded using Bayes rule. But:
$$ p(w \mid X, Y) = \frac{p(X, Y \mid w) \pi(w)}{p(X, Y)} = \frac{p(X \mid w) p(Y \mid X, w) \pi(w)}{p(Y \mid X) p(X)}.$$
So it seems like the development in the paper is assuming $p(X \mid w) = p(X)$. Can the authors correct what I've written here or explain why this assumption is true?
6. "samples from $\hat\rho$ are inputted to the full neural network." I thought that $\hat\rho$ was a distribution over neural networks. What does this sentence mean?
7. In figure 3 and 4, ten blue lines are shown, one for each posterior. But only one red line (PAC-Bayesian bound) is shown. Why are there not ten red lines, which each one showing the PAC-Bayesian bound for each posterior? Is the displayed red line the mean of these ten lines?
8. In Section 4, $\phi_i$ is used, but I don't think this was ever defined.

**Strength And Weaknesses:**

I have a few issues with the paper as-is; I've broken these down into categories below.

**Discussion of previous work.** I think the discussion of previous work is flawed in a few places.
1. The problem this paper studies is mathematically equivalent to tempering the likelihood in Bayesian inference. Maybe the literature has advanced and I've missed something, but I don't think this is the setting people talk about when they discuss the "cold posterior effect," which is what the current paper says it is about. For example, the authors write "As discussed in ... Wenzel et al. (2020) the relevant setting for the ELBO is the one we consider..." I don't think Wenzel et al. (2020) says this. That paper notes that people have considered tempering just the likelihood in the context in variational inference, but they differentiate this likelihood-tempered setting from the cold posterior setting they consider in Appendix E.
1. Citations of theorems from other papers (Theorems 1 and 2) should explicitly cite the theorem from that work. This is especially important for something like Catoni, 2007, which is 175 pages long; as a reader, there's no practical way for me to find the relevant theorem from that book. Also, have all the conditions for each theorem been recalled here? Theorem 1 looks like Theorem 4.1 from Alquier et al. 2016, which has an assumption on it that isn't recorded in the current paper.
5. The paper is written as if it is about Bayesian inference. But in the experiments, the authors note "We first use $Z_{train}$ [the training data] to find a prior mean $w_\pi$." This means the experiments are about an *empirical Bayesian* method; Empirical Bayes is not the same thing as standard Bayesian inference.
2. "...the Blackwell-Dubins consistency theroem..." I think Doob's theorem is a much more common reference for the contraction of Bayesian posteriors.  But it's important to note that posterior asymptotics are going to be different when using regular Bayesian inference than when using an Empirical Bayesian procedure (which is what this paper seems to be about), and results for one method may not be valid for the other.
3. "Germain et al. (2016) were the first to find connections between PAC-Bayes and Bayesian inference." I don't think this is quite true -- e.g. Grunwald (2012) cited here discusses PAC-Bayes. Is there some more specific context in which this is true?

**Theoretical development (Section 4).** I'm not sure what the takeaways from the authors' theory is, given some issues with the assumptions going into it and the discussion of their results.
1. Why is taking $\sigma = 1$ a reasonable model? Is there anything lost by making this assumption?
2. It is assumed that the gradients of a neural network come from a Gaussian mixture, and the fact that previous work has shown neural network gradients to be clusterable (Zancato et al. 2020) is cited as justification. I don't see why the results of Zancato et al. (2020) imply that the gradients should come from a normal distribution -- data can be clusterable and not come from a mixture of Gaussians. For example, a mixture of uniform distributions will be very different from a mixture of Gaussians.
3. The proved PAC-Bayesian bound requires the temperature $\lambda$ to be $O(1/n)$, where $n$ is the number of training datapoints. But "cold posterior" effects will show up for large $\lambda$. So what interesting behavior can we extract from this bound? E.g. for $\lambda \in (0, 0.0001)$ (which this bound is restricted to for even moderate $n$), all of the plots in the paper show the same behavior.
4. There isn't much discussion of the proved bound except that it displays "complex interactions" between the model and data. Some more discussion seems needed here. Can we learn anything from these complex interactions? What do the interactions tell us about the cold posterior effect? I think it's clear that the assumptions going into the bound are unrealistic. But if the bound for this toy model matches empirical results in interesting ways that previous bounds do not, this would still be a great contribution to the literature.

**Empirical results.** I wasn't entirely clear on what the takeaway was supposed to be from the experiments. These seem to be evaluating existing PAC-Bayesian bounds; overall, what does each experiment tell us about understanding the cold posterior effect? It seems to me that most of the experiments just validate the cold posterior effect by showing larger $\lambda$ gives better performance. Is there something that I'm missing here? Also a few more specific points:
1. "We might think these results are due to the poor (Isotropic) approximation to the posterior, however as we will see in the next section this behaviour carries over to other approximations to the posterior." This next section changes both the model (regression -> classification), the bound used (Theorem 1 -> Theorem 2), and the posterior approximation. Too much is different here to validate that changing the approximation isn't changing things; how do we know that the effect of changing the approximation isn't being canceled out by one of the other changes?



**Summary Of The Paper:**

This paper studies what happens in Bayesian deep learning if the likelihood is scaled down by a factor of $1/\lambda$. This is related to the "cold posterior effect". The authors empirically show that two different PAC-Bayesian bounds, which also include a scaling of $1/\lambda$, correlate well with this tempered posterior's performance. Finally, the authors derive a PAC-Bayesian bound for a simplified model neural network and show that it has a complex dependence on the temperature $\lambda$.

**Summary Of The Review:**

Overall, I think the paper needs more connection with the literature, and also a deeper explanation of the presented results. As-is, it's not clear to me what the new high-level takeaways are from this paper.

---

> ### Author Response · Authors · 2022-11-06
> **Reply qVvM**
>
> Thank you for your detailed review!
>
> Discussion of previous work:
> 1) In [2] p2. the authors discuss how for Gaussian priors the scaling factor (1/T) can be absorbed in the prior variance, thus making the cold and tempered objective equivalent for a rescaling of the prior variance. From Appendix E in [1] we see that tempering the KL is equivalent to tempering only the likelihood which is equivalent to tempering both likelihood and prior in the Gaussian case. We can add this point in our F.A.Q.
>
> 2) For the Alquier theorem we are using the version found in [3] p4 Theorem 3, which is copied as it was. For the Catoni bound we use Theorem 1.2.6 from [4]. We'd be happy to provide specific answers for other theorems and include the Theorem numbers in the revised version.
>
> 3) This is a fine point. We use this approach (learning a prior using the training set and a posterior partially using the validation set) only when estimating the PAC-Bayes bounds, to make the bounds tight. In Figure 3 middle and bottom rows as well as in Figure 4, the Laplace approximation is estimated using the standard approach of starting from some random initialization finding a MAP then fitting the Laplace all using the training set. As we understand it, this is a valid standard Bayesian inference approach.
>
> 4) We believe that our response in q3 also holds for this question. We are happy to use the second reference.
>
> 5) We agree that the language here is vague. [3] were the first to point out that minimizing a PAC-Bayesian generalization risk bound
> maximizes the Bayesian marginal likelihood. In [5] the authors propose the Cèsaro average of Bayesian posteriors which the show exhibits better properties than the standard posterior through a PAC-Bayesian analysis. We can make this clear in the main text.
>
> Theoretical development
>
> 1) We discuss in the main text that finding the prior variance through the Marginal Likelihood results in to small of a variance which is not relevant for our experiments. We can only comment that for other non-trivial prior variances the behaviour we observed was similar, and that doing a grid search (as we would also prefer) would be very costly. In the middle and bottom rows of Figure 3 as well as in Figure 4 we learn the prior variance using the Marginal Likelihood and the intuition we derive still holds.
>
> 2)We agree that this is strong assumption, but one that is necessary to get something workable in the bound. In any case we only expect some very rough intuition from this approach.
>
> 3) It is not necessary the case that \lambda is O(1/n) as \sigma_{\pi}^2 and or \sigma_x^2 could be small. However we again agree that the bound is a very coarse approximation.
>
> 4)What more can we expect to find from Proposition 1? One interesting direction would be to tease apart the influence of the parameters that appear in the bound such as h, sigma_{\pi}^2, w_* etc. We believe that this is an interesting direction for future work, as it requires a significant amount of new experiments, in an already long and involved paper.
>
> Empirical results
>
> Our results show something stronger than the cold-posterior effect. In [1] the authors claim that some stochasticity comes with benefits for the predictor. This is a very suspicious claim, for example in [1] p1 Figure 1 for the coldest temperature the deterministic and stochastic accuracies don't match! One can make a number of guesses as to why this happens for MCMC, but analyzing the MCMC case is inherently difficult. By using the Laplace approximation we make the inference problem considerably simpler and controllable. By definition now \lambda -> \inf in the isotropic case equals the MAP. We empirically show that being even a bit Bayesian always hurts accuracy.
>
> We then switch to the NLL. We show here much more pronounced and diverse effects, which differ from the effects on the misclassification rate. This hasn't been adequately discussed before in the literature. In fact [6] show results only for cross entropy in the main text, which exaggerates the perceived impact of stochasticity and the cold posterior effect in general. Looking at the Appendix of [6] any change in test accuracy for \lambda>1 is of the order of 1% and often lower. We highlight that conclusions about the cold posterior effect can vary alot depending on the chosen metric.
>
> Motivated by the results in NLL we ask the more pertinent question that hasn't been asked in previous cold posterior papers: "Given that different test metrics behave differently for different temperatures can we find a temperature that improves all metrics?". In the Appendix we derive Pareto fronts of test accuracy vs test ECE, we see that we can't improve the ECE without hurting the accuracy.

---

> > ### Author Response · Authors · 2022-11-06
> > **Reply qVvM (continued)**
> >
> > Empirical results (Continued)
> >
> > All in all previous works paint an incoherent picture of the cold posterior effect, seeming to both criticize standard Bayesian inference while claiming that some stochasticity helps. Our results are more coherent. Concerning the primary classification metric (the misclassification rate) stochasticity always hurts performance. For secondary metrics, we can't get benefits without sacrificing accuracy. The PAC-Bayes bound interpretation also closely tracks the primary metric and gives the correct intuition that being Bayesian always hurts the primary metric.
> >
> > --"We might think these results are due to the poor ..." In Figure 3 we plot the Catoni bound, the standard Isotropic Laplace, as well as the K-FAC Laplace. Most other settings are the same across these experiments. Therefore we can directly compare these three cases together.
> >
> > Clarity, Quality, Novelty And Reproducibility:
> >
> > 1)We believe (X,Y)\sim \mathcal{D}^n is correct as (X,Y)\in(\mathcal{X}\times\mathcal{Y})^n
> >
> > 2-4) We will fix these points in the revision
> >
> > 5) X are the inputs to the model. As such p(X|\bw)=p(X).
> >
> > 6) \hat{\rho} is better thought of as the distribution over the neural network weights. We wrote this sentence only because there exist works that sample from the K-FAC Laplace posterior and then use a linearized version of the neural network to make predictions. We instead use the full network. It's weight follow \hat{\rho}.
> >
> > 7) The solid line is the mean of the bounds. We felt that the plotting the samples as well would clot even more the Figure but can change that in the revision.
> >
> > 8) \phi_i should be positive constants that sum to 1. We will fix this point in the revision.
> >
> >
> >
> >
> >
> > [1]: Wenzel, Florian, et al. "How good is the bayes posterior in deep neural networks really?." arXiv preprint arXiv:2002.02405 (2020).
> >
> > [2]: Aitchison, Laurence. "A statistical theory of cold posteriors in deep neural networks." arXiv preprint arXiv:2008.05912 (2020).
> >
> > [3]: Germain, Pascal, et al. "PAC-Bayesian theory meets Bayesian inference." Advances in Neural Information Processing Systems 29 (2016).
> >
> > [4]: Catoni, Olivier. "PAC-Bayesian supervised classification: the thermodynamics of statistical learning." arXiv preprint arXiv:0712.0248 (2007).
> >
> > [5]: Grünwald, Peter. "The safe bayesian." International Conference on Algorithmic Learning Theory. Springer, Berlin, Heidelberg, 2012.
> >
> > [6]: Noci, Lorenzo, et al. "Disentangling the roles of curation, data-augmentation and the prior in the cold posterior effect." Advances in Neural Information Processing Systems 34 (2021): 12738-12748.

---

> > > ### Comment · Reviewer_qVvM · 2022-11-11
> > > **Thanks for the replies!**
> > >
> > > Thanks a lot for the responses. I'm still hung up on a couple points, and am hoping we can continue the conversation here. I'm using the same numbering / sections below that you've used above:
> > >
> > > **Discussion of previous work:**
> > >
> > > 1. I see the mathematical equivalence of the cold and tempered posteriors as discussed on p.2 of [2]. But I think these are only equivalent for a fixed temperature, as the tempered posterior has a prior variance that is fixed with respect to the temperature. So, while the tempered and cold can be made equivalent for a given $\lambda$, it doesn't seem like they are equivalent for all $\lambda$. Since the current paper is studying the effect of varying $\lambda$, it seems like this is an important difference.
> > > 2. Thanks for the reference. Theorem 1 from [3] does not look like Theorem 4.1 from Alquier et al. (2016) (it's removed an assumption), so I think it's important to cite the actual version of the Theorem used. Given that [3] goes into a lot of detail about how to ensure the moment term is bounded, I really agree with reviewer oDKT that the boundedness of this term is concerning. I'm wondering if the authors can provide some detailed discussion of how they would rewrite things to discuss this issue. I see that this term is explicitly computed in Proposition 1, but presumably the authors want their results to apply beyond the assumptions that have gone into Proposition 1.
> > > 3. This makes sense to me. It might be good to have some language clarifying which distribution $\pi$ is just for evaluating the PAC bound and which is the actual prior used to compute the posterior.
> > >
> > > **Theoretical development**
> > > \
> > > (2) and (4) I think it's totally fair to use coarse approximations to get theoretical developments. But I think those approximations then have to be connected back to reality. In particular, I don't think this theorem says much without some discussion that does two things: (A) tells us how the theorem gives us new predictions / insights / explains behavior that was formerly unexplained  *in the setting where the assumptions hold* and (B) justifies why we should still expect the results from (A) to hold even when the assumptions are significantly violated in real life. An alternative to (B) would be to argue that the assumptions aren't significantly in real life, but I think they are significantly violated in real life.
> > >
> > > (3) So does this theorem only tell us useful things when $\sigma_x$ and/or $\sigma_\pi$ are asymptotically vanishing? Can the authors justify why this is a regime that we should care about / expect to see in practice? Again, this connects back to my points in (2) / (4), which is trying to get at what this theorem tells us about real life situations.
> > >
> > > **Experiments**
> > > I'm not sure what the authors have written here addresses the fact that the experiments in Section 3.2 don't address the concern that the results from Section 3.1 are just because of the "poor (Isotropic) approximation to the posterior". But, I do think what the authors have written here is a much more directly compelling empirical story than what is currently written in the paper. I think this new discussion puts the empirical results in a really interesting light that others might learn from! But then I'm not exactly sure how this new story fits in with the rest of the paper (especially the theoretical developments).
> > >
> > > Overall, I agree with reviewer oDKT, who writes "I think it isn't totally clear what the take-away of the paper is".
> > >
> > > **Clarity, Quality, Novelty And Reproducibility:**
> > > \
> > > (7) I don't think it's necessary to show all the red lines in the main text. But what the red line is should definitely be explained (I don't *think* I saw it in the main text, definitely not in the caption).

---

> > > > ### Author Response · Authors · 2022-11-17
> > > > **Reply 2!**
> > > >
> > > > **Discussions of previous work:**
> > > >
> > > > 1) We agree on this point. One can make the argument that taking a grid search over both \lambda and the prior variance for both tempered and cold objectives should make the tempered posterior informative about the minima of the cold one. However, please note that we didn't wish to make such a claim (of equivalence) in the original submission (though the language in that section is somewhat vague), but merely to note that [1] discusses the tempered objective. Recent works such as [5] p6 (ICLR 2021), [6] p6 (NeurIPS 2022), [7] p3 (ICML 2022) discuss and conduct experiments on the tempered objective making our work highly relevant.
> > > >
> > > > 2) We propose to add to the text that we are using the version of the theorem found in [2][3], which was however originally proposed in [4]. The Hoeffding assumption in [4] is not strictly necessary for the bound. However this assumption, or an assumption such that the random variable $V=\mathcal{L}_{\mathcal{D}}^{\ell}(f)-\ell(f,x,y)$ is sub-Gaussian, sub-gamma or sub-Weibull is necessary to ensure that the Moment term is bounded. We propose to add this discussion to the text, in light of the fact that we don't regard this assumption to be too restrictive and these assumptions have been discussed before in the literature [2,4]. Furthermore, we intend to move the entire section on the regression experiment to the Appendix as it is of smaller value than an extended discussion of the classification experiments. For the classification case, the loss is bounded and therefore the Moment term is de facto bounded as well.
> > > >
> > > > **Theoretical development**
> > > >
> > > > We believe that we can derive a simple, and novel intuition from our bound that can also be observed in our experimental data. Specifically, for fixed empirical risk, temperature, prior variance, and assuming a fixed labelling function, our bound still implies variability in the test accuracy simply because of the properties of the different MAP estimates, specifically the curvature $h$ and distance from initialization $\Vert w_{\hat{\rho}}-w_{\pi}\Vert_2^2$. Indeed for some fixed temperatures, we can observe a large variability of order $O(10)$ (in percentage points of accuracy), while the variability from the MCMC approximation of the predictive should be of order $O(1)$. This directly contradicts previous work based on data augmentation, misspecified likelihood and prior, as all these are fixed (and data augmentation isn't enabled in the main experiments). The same temperature should result in the same test accuracy. It furthermore implies that it might be difficult to find a single value of $lambda$ that is always optimal (as suggested by previous works), even for the same dataset, as inference is always approximate even in MCMC. We thank you for triggering this novel intuition which we are incorporating in the revised version.
> > > >
> > > > **Experiments**
> > > >
> > > > We agree that sections 3.1 and 3.2 are not directly comparable. As mentioned above we will move the entire regression section to the Appendix, as we believe that the classification results are the most relevant for the community. We will also modify the text there such that we remove the claim that results from section 3.2 transfer to section 3.1.
> > > >
> > > > Thank you for your comments on our experiments highlighting which perspective is relevant for the community. We will adapt the organization of the text to reflect this discussion.
> > > >
> > > > **"I think it isn't totally clear what the take-away of the paper is"**
> > > >
> > > > We will improve the motivation of our approach in the Introduction, making clear the following take-aways:
> > > >
> > > > 1) We first discuss that optimizing purely Bayesian objectives such as the ELBO might be ill-suited when targeting Frequentist metrics such as test accuracy. This might motivate practitioners to either target more traditional Bayesian metrics such as contraction to the true posterior, or use heuristics such as tempering "guilt-free" when targeting Frequentist metrics.
> > > >
> > > > 2) Empirically we show that in most cases Bayesian inference using the Laplace approximation doesn't improve the primary Frequentist metric (test accuracy) at any temperature $\lambda$. This is a stronger and more coherent observation than the one in [1]. We show that PAC-Bayes bounds correlate with this observation. Secondary metrics have a more complex relationship to $\lambda$. We show that we cannot improve them without impairing the primary metric.
> > > >
> > > > 3) See theoretical development above.
> > > >
> > > > All the above paint a more coherent picture compared to previous works, all under the perspective of PAC-Bayes.

---

> > > > > ### Author Response · Authors · 2022-11-17
> > > > > **cited**
> > > > >
> > > > > [1]:Wenzel, Florian, et al. "How good is the Bayes posterior in deep neural networks really?." arXiv preprint arXiv:2002.02405 (2020).
> > > > >
> > > > > [2]:Masegosa, Andres. "Learning under model misspecification: Applications to variational and ensemble methods." Advances in Neural Information Processing Systems 33 (2020): 5479-5491.
> > > > >
> > > > > [3]:Germain, Pascal, et al. "PAC-Bayesian theory meets Bayesian inference." Advances in Neural Information Processing Systems 29 (2016).
> > > > >
> > > > > [4]:Alquier, Pierre, James Ridgway, and Nicolas Chopin. "On the properties of variational approximations of Gibbs posteriors." The Journal of Machine Learning Research 17.1 (2016): 8374-8414.
> > > > >
> > > > > [5]:Aitchison, Laurence. "A statistical theory of cold posteriors in deep neural networks." International Conference on Learning Representations (2021).
> > > > >
> > > > > [6]:Kapoor, Sanyam, et al. "On Uncertainty, Tempering, and Data Augmentation in Bayesian Classification." Advances in Neural Information Processing Systems (2022).
> > > > >
> > > > > [7]:Bachmann, Gregor, Lorenzo Noci, and Thomas Hofmann. "How Tempering Fixes Data Augmentation in Bayesian Neural Networks." International Conference on Machine Learning    (2022).

---

### Official Review · Reviewer_oDKT · 2022-10-23

**Confidence:** 3
**Correctness:** 3
**Technical Novelty And Significance:** 3
**Empirical Novelty And Significance:** 3
**Recommendation:** 5

**Clarity, Quality, Novelty And Reproducibility:**

## Quality and Reproducibility
- Experiments appear to be thorough. The appendix includes details.
- Monte Carlo estimation of the moment term seems problematic. In particular, the moment term roughly corresponds to a condition insisting that the tails of a distribution are light enough, and therefore could be very high variance or infinite. I am not convinced 100 MC samples would necessary be reasonable for estimating this, without some knowledge of the tail behavior (e.g. this would be quite different in the case of a bounded random variable, where a concentration inequality could be applied). The authors mention this briefly in the supplement, but in a way that suggests that since 100 samples suffice for estimating some statistics of the posterior, they would also be sufficient for estimating this term. However, since the term involves the exponetial of the random variable in question, this seems speculative.

## Clarity
- Aspects of the combination of Bayesian and frequentist epistemology in the introduction are confusing. For example, the authors argue (from the frequentist perspective) that a Bayesian approach maybe be suboptimal. Then state "the posterior is our best guess at the true model parameters without having to resort to heuristics". As far as I know, this is only true from a (subjective) Bayesian view, as a frequentist might prefer, for example, a minimax approach. The authors should be more clear about what assumptions are being made and consistent about the view that is taken or delineate clearly when the viewpoint changes.
- A better summary of the experiments section would be helpful. There is a lot going on, which is great. But perhaps a table clarifying what parameters were selected using what method and what data (perhaps in the supplement) would improve readability.

## Questions

- Is it clear that the "moment term" is even finite for $\lambda >0$ in the application considered?
- It may be worth mentioning explicitly that the empirical version of minimizing the divergence in equation 2 is maximum likelihood.
- Can you clarify how the MAP estimate can lead to the optimal bound? I don't see how this can be the case unless you have a prior variance that is also $0$, in which case the whole method must ignore the validation data and is equivalent to using a validation set-based bound. Otherwise, the Kullback-Leibler divergence should tend to infinity leading to a vacuous bound.
- Is the PAC-Bays posterior selected based both on training and validation data (as is standard)? I suspect I have missed this in the appendix despite looking for it, and would appreciate a pointer.

## typos
- p1. ``tells states''


**Strength And Weaknesses:**

## Strengths

- The writing is clear; figures are high quality.
- Experiments seem reasonably thorough.

## Weaknesses

- I found it hard to pin down exactly what insight about (Bayesian) deep learning is gained. The authors show that lower temperatures correspond correlate with better generalization and tighter generalization bounds. I would like the authors to clarify what this correlation teaches us about deep learning above and beyond what is known. The claim that a tighter generalization bound correlates with better generalization does not seem inherently interesting, and I think needs to be tied more clearly to concrete/actionable insight.

**Summary Of The Paper:**

The authors consider the application of PAC-Bayes theory to study the "cold posterior effect" previously observed in Bayesian neural networks. They rely on the aesthetic relationship between cold posteriors and certain PAC-Bayes bounds on the negative log likelihood to carry-out their analysis. They consider the negative log-likelihood loss for regression and the $0-1$ loss for classification. The authors rely on data-dependent priors selected with a validation set, then apply a Laplace approximation to determine the posterior (in the PAC-Bayes sense). From the experiments, the authors conclude that cold posteriors often lead to better generalization bounds, and suggest this as a mechanism for explaining the cold posterior effect.

**Summary Of The Review:**

I think this paper is slightly below the acceptance threshold. The writing is good, experiments are thorough and well-thought out. However, I think some details of the experiments (e.g. Monte Carlo estimation of the moment term) lead to technical issues; and it is somewhat problematic that the generalization bounds reported may not be reflective of the actual values due to sampling error. More importantly, I think it isn't totally clear what the take-away of the paper is, in terms of insight that sheds more light on deep learning/Bayesian deep learning. The writing should make this evident. The discussion section points to more things that could be done, but really I think this space would be better spent clarifying how what was done in this paper relates to the bigger picture. It is quite possible I have missed something regarding this, and I hope the authors can clarify this in their response.

---

> ### Author Response · Authors · 2022-11-07
> **Reply oDKT**
>
> Thank you for your detailed review!
>
> The main take-aways from our work are the following:
>
> [1] Has been used as a poster-child for supposed "inherent problems" with Bayesian deep learning. Therefore, considerable effort has been put into refuting [1]. We ask the question: "Is it even correct to try to compete at Frequentist metrics with *purely* Bayesian approaches (such as the ELBO) when these have been traditionally used to infer the correct model posterior?" We argue that it might not be correct, which might save time for Bayesian deep learning practitioners. It might motivate them to use heuristics such as tempering "guilt-free" or alternatively evaluate on different metrics such as posterior contraction which are better founded for purely Bayesian approaches. We believe that this is not a small problem, as as a whole the Bayesian DL has been chasing Frequentist metrics with very mixed results.
>
> For the case of the K-FAC Laplace approximation we have furthermore shown empirically that being even a bit Bayesian does not help with any of the most popular Frequentist evaluation metrics (at least without hurting another). This is a stronger result than the original cold posterior paper [1], which supposedly shows small improvements for some stochasticity. Note that the results in [1] are very suspicious, in [1] Figure 1 for the coldest temperature the Bayesian and deterministic performance doesn't match. By contrast, in our Laplace case by definition the deterministic and coldest temperature match. It's clear that even tempering doesn't seem to work, which poses the important question of how can we make the Laplace approximation consistently improve upon the MAP. This is again in contrast to previous literature [2] which implies that some stochasticity helps. Our (Catoni) PAC-Bayes bound interpretation correctly tracks this behaviour.
>
>
>
> Answers to specific questions:
> 1) To show that the Moment term is upper bounded, one needs some assumptions on the loss. For example one can assume that the loss is a subgaussian random variable under the prior \pi and the input distribution \mathcal{D}. The assumptions that we make for the labelling function in page 8 implicitly result in a bound on the Moment term. In the general case of unbounded losses (where we can't model the labelling function, such as in Figure 2) we believe that the sub gaussianity assumption is plausible. Note that in the classification setting, Figures 3 and 4 (which is the most important) the Moment term is bounded as the 0-1 loss is bounded. As such in the classification setting there is no question of numerical stability. This is also reflected in the Catoni bound (Theorem 2). See [3] p4 for a more detailed discussion on the Moment term.
> 2) Yes we will add this in the final version.
> 3) While the KL term increases as \lambda -> inf note that in the bound we divide by \lambda such that the contribution is (1/\lambda)*KL(\hat{\rho}||\pi), therefore as \lambda-> inf the total contribution to the bound actually diminishes.
> 4) For the PAC-Bayes case we find the posterior (and prior) mean using the training set. We then find the posterior variance using the validation set. We also evaluate the bound using the validation set. Thus the posterior is learned jointly in a simplified online fashion through both the training and validation sets. This is described in pages 6 and 7 of the main text as well as page 7 top paragraph, but we will try to make it more clear in the revision.
>
> [1]: Wenzel, Florian, et al. "How good is the bayes posterior in deep neural networks really?." arXiv preprint arXiv:2002.02405 (2020).
> [2]: Daxberger, Erik, et al. "Laplace redux-effortless bayesian deep learning." Advances in Neural Information Processing Systems 34 (2021): 20089-20103.
> [3]: Germain, Pascal, et al. "PAC-Bayesian theory meets Bayesian inference." Advances in Neural Information Processing Systems 29 (2016).

---

> > ### Comment · Reviewer_oDKT · 2022-11-17
> > **Follow up on estimating the moment term**
> >
> > Thanks for the response and for clarifying the effect of increasing lambda in point 3 above. If I understand correctly, the effect is essentiallly to decrease the "cost" of having a high KL divergernce to the prior with a trade-off in having a larger "moment term" (in the Alquier bound at least, it appears to have a different effect in the Catoni bound).
> >
> > This does make me worried about the validity of experiments in cases where the loss isn't bounded (or provably subGaussian) as it would seem to me that the moment term is going to be very difficult to estimate for large $lambda$. I am still not at all convinced that estimating this term with Monte Carlo is reasonable, and I would expect the variance of the estimator to depend on the magnitude of lambda. Even in the case of bounded random variables, I wouldn't expect to be able to estimate log E[exp(lambda X)) via Monte Carlo for lambda~10^4, as log E[exp(lambda X)] will be controlled almost entirely by the most extreme values of X, which may not be observed at all when using Monte Carlo with any reasonable number of samples. I read the section you point to in [3] above, but didn't find anything that would suggesting estimating the moment term via Monte Carlo, particularly for large lambda, is reasonable. Any more thoughts on this?

---

> > > ### Author Response · Authors · 2022-11-17
> > > **Estimating the moment term.**
> > >
> > > We believe that there might be a misunderstanding on this point. For the bounded loss case (specifically for the 0-1 loss) the Catoni bound incorporates a direct upper bound to the Moment term. As such we never have to estimate the Moment term for this bound. This can be seen from the form of the Catoni bound in page 5. Alternatively one can see that this is the case from the way the Catoni bound is derived (see [1] pages 14 to 15 "Basic inequality"). The Moment term is absorbed in the function \Phi appearing in the bound. Therefore there are no possible numerical or convergence issues for our most important experiments, the classification experiments on large ResNets. (Importantly for the classification setting we are only using the Catoni bound. Bounding the NLL is prohibitive in general, for many reasons apart from the ones mentioned by the reviewer.)
> > >
> > > We consider the regression experiments of much smaller importance and will be moving them to the Appendix so as to highlight other experiments on the classification setting. Here the reviewer has raised some interesting points. Even for random variables with light tails, our estimator for the log-Laplace transform could have high variance. We can only comment that the estimate is probably unreliable for large values of lambda, and for the Naive Monte Carlo estimator employed here. Furthermore, the design of better estimators is an ongoing topic of research [2].
> > >
> > >
> > > [1] Catoni, Olivier. "PAC-Bayesian supervised classification: the thermodynamics of statistical learning." arXiv preprint arXiv:0712.0248 (2007).
> > > [2] Asmussen, Søren, Jens Ledet Jensen, and Leonardo Rojas-Nandayapa. "On the Laplace transform of the lognormal distribution." Methodology and Computing in Applied Probability 18.2 (2016): 441-458.

---

### Official Review · Reviewer_Pea3 · 2022-10-25

**Confidence:** 4
**Correctness:** 4
**Technical Novelty And Significance:** 2
**Empirical Novelty And Significance:** 2
**Recommendation:** 5

**Clarity, Quality, Novelty And Reproducibility:**

The paper is generally well written and all the results are presented and the results appearcorrect.

In terms of novelty, the authors study various previously studied approximation to the Hessian and apply those in the context of Laplace approximation of the posterior and thus I consider it to be incremental progress.

In terms of reproducibility, I see some of the PAC-Bayes bounds appear to be smaller than the test error obtained. Authors should comment on that because PAC-Bayesian bounds ought to be an upper bound.

**Strength And Weaknesses:**

## Strengths

The paper is generally well organized.

Paper tries to study Bayesian phenomenon of cold-posterior from a frequentist point of view of PAC-Bayesian bounds. deriving performance guarantees of an approximation of the Bayesian posterior.

## Weaknesses

- In my opinion, the problem of studying the cold-posterior on itself is not well motivated as Bayesian analysis is more useful for uncertainty quantification rather than pure prediction. Of course, one can improve prediction quality if the posteriors are sharper.
- For the linearized network, authors merely suggest a connection to the cold-posterior without attempting to tease apart details.

**Summary Of The Paper:**

In this paper authors do a PAC-Bayesian analysis for the cold-posterior effect, i.e. an observation that sharp posterior typically lead to better predictive performance. Main contribution of the paper is obtaining a PAC-Bayesian bound for Laplace approximation to the posterior. Since Laplace approximation relies on the Hessian, authors study versions of generalized Gaussian—Newton approximations to the posterior. Under these approximations, authors experimentally show that “colder” posterior yields better smaller PAC-Bayesian bound. Authors also derive PAC-Bayesian bound for stochastic linearlized network.

**Summary Of The Review:**

I recommend a weak reject for the paper. Main reasons for my recommendation is that the problem being studied is not very well motivated.  And the fact the technical developments are only incremental progress.

---

> ### Author Response · Authors · 2022-11-06
> **Reply Pea3**
>
> Thank you for your detailed review!
>
> The reviewer mentions that "the problem of studying the cold-posterior on itself is not well motivated as Bayesian analysis is more useful for uncertainty quantification rather than pure prediction".
>
> **In our work we show that not only the Laplace approximation always hurts accuracy but most importantly it does not improve the ECE or the NLL without hurting accuracy, for any temperature \lambda. We thus urge the reviewer to reevaluate the impact of our work.**
>
> 1)Most current Bayesian DL practitioners [1][2][3][4][5] benchmark on Frequentist metrics and specifically on test ACC/NLL/ECE with very mixed results. Also considerable effort has been put in refuting the cold-posterior effect [6][7][8].
>
> We ask the question "Should practicioners try to beat Frequentist metrics using purely Bayesian approaches?" We propose that this might be ill founded, as in our case the ELBO doesn't bound the quantity of interest. Thus practicioners should either use heuristics "without guilt" when benchmarking on Frequentist metrics, or they should focus on fidelity to the true posterior which is much better posed for purely Bayesian approaches.
>
> 2)In the main paper together with the *Appendix* we show empirically that being Bayesian with the K-FAC Laplace, doesn't improve any metric *even the Expected Calibration Error (ECE)* (at least without hurting another). In the ECE case which the reviewer might consider the most important (and we do to), improving the ECE results in degrading test ACC. This begs the important question: Can we improve the Laplace approximation such that it always improves upon the MAP? Previous work [4][5] obscures this point.
>
> **The reviewer can find the ECE experiments in the Appendix. However we will also move them in the main text in the revision.**
>
>
>
> [1]: Wenzel, Florian, et al. "How good is the bayes posterior in deep neural networks really?." arXiv preprint arXiv:2002.02405 (2020).
>
> [2]: Izmailov, Pavel, et al. "What are Bayesian neural network posteriors really like?." International conference on machine learning. PMLR, 2021.
>
> [3]: Nalisnick, Eric, Jonathan Gordon, and José Miguel Hernández-Lobato. "Predictive complexity priors." International Conference on Artificial Intelligence and Statistics. PMLR, 2021.
>
> [4]: Daxberger, Erik, et al. "Bayesian deep learning via subnetwork inference." International Conference on Machine Learning. PMLR, 2021.
>
> [5]: Immer, Alexander, Maciej Korzepa, and Matthias Bauer. "Improving predictions of Bayesian neural nets via local linearization." International Conference on Artificial Intelligence and Statistics. PMLR, 2021.
>
> [6]: Noci, Lorenzo, et al. "Disentangling the roles of curation, data-augmentation and the prior in the cold posterior effect." Advances in Neural Information Processing Systems 34 (2021): 12738-12748.
>
> [7]: Bachmann, Gregor, Lorenzo Noci, and Thomas Hofmann. "How Tempering Fixes Data Augmentation in Bayesian Neural Networks." arXiv preprint arXiv:2205.13900 (2022).
>
> [8]: Aitchison, Laurence. "A statistical theory of cold posteriors in deep neural networks." arXiv preprint arXiv:2008.05912 (2020).

---

### Author Response · Authors · 2022-11-18
**Summary of revisions**

We would like to thank all the reviewers, for their detailed, thoughtful, and useful comments.

**Revisions**
We have made a number of revisions to the main text following the discussion period.

1) We moved the regression experiments to the Appendix in order to make room for more relevant classification experiments.
2) We moved the experiments which plot Pareto fronts of the ECE compared to the 0-1 test risk, to the main text. These show that not only does the Laplace hurt accuracy for almost all temperatures, but also for all temperatures there is a tradeoff between ECE and accuracy. As such one cannot in general improve calibration without hurting accuracy. This result is novel and might answer the concerns of Reviewer Pea3.
3) We made new experiments based on Proposition 1. These show that for different MAP estimates and their corresponding Laplace approximations the test accuracy fluctuates significantly for fixed $\lambda$, a fixed prior and a fixed likelihood. Alternatively, to achieve a target test accuracy for a fixed prior and a fixed likelihood the required $\lambda$ fluctuates with the different MAP estimates (even though they have the same training accuracy). These results are novel, and highlight that it might be difficult to find a single $\lambda$ corresponding to some target aleatoric uncertainty (as suggested by previous works) that works for any run of an approximate inference procedure.

Finally following the discussion period we are also open to changing the title to "Tempered posteriors through PAC-Bayes", which might better reflect the content of the submission.

**Insights**

Some reviewer felt that the message of the paper is unclear. We believe that our PAC-Bayesian viewpoint gives a number of insights:

1)We argued that standard Bayesian inference does not readily provide high probability bounds on test metrics, yet is commonly used to target such metrics. We hope to motivate Bayesian researchers to either accept heuristics when targeting Frequentist metrics or target posterior contraction which is much better posed for standard Bayesian inference. This might save considerable research resources. As an example, the cold posterior effect fits naturally with a proper PAC-Bayes bound on test 0-1 loss, as opposed to a misapplied ELBO.

2)We show experimentally that the Laplace approximation does not provide any clear advantages over the MAP, neither in test accuracy, nor more importantly in calibration. The Catoni PAC-Bayes bound correlates well with the test accuracy for different $\lambda$.

3)Our PAC-Bayesian analysis furthermore implies (and we confirm experimentally) that for a target test 0-1 loss the required $\lambda$ changes according to the MAP of the Laplace. This contradicts previous analyses, that relate $\lambda$ to aleatoric uncertainty, a quantity that remains unchanged for different minima of the loss.

---

### Decision · Program_Chairs · 2023-01-20

**Decision:**

Reject

**Justification For Why Not Higher Score:**

The take-home message is unclear, as acknowledge by the authors in their "meta-rebuttal", saying they would rename the paper, better lay out the messaging, move some part to the appendix, etc.


**Justification For Why Not Lower Score:**

The contribution has potential. After a few rounds of rewriting, pruning, and editorial positioning, the material presented here could make for a good paper.

**Metareview: Summary, Strengths And Weaknesses:**

This paper provides a PAC-Bayesian analysis for the cold-posterior effect, that is the observation that in (Deep) Bayesian inference approaches, sharp posterior typically lead to better predictive performance -- which might be counterintuitive and/or depart from the fully Bayesian treatment of the learning task and go towards championing more frequentist approaches. The work builds takes inspiration from a connection between PAC-Bayesian bounds and the ELBO (Evidence Lower Bound), the critical quantity optimized over in Bayesian inference.

+ the writing is clear, with efforts to make the equations more readable and relatable to each other and the text
+ the empirical evaluation is thorough
- the message of the paper is not clear: what is the take away, what is the contribution made here? what are the lessons learned?
- (a remark from AC) PAC-Bayesian bounds are designed for the risk of the stochastic Gibbs classifier, and some gymnastics is needed for those bounds to apply to deterministic classifiers; here, the Laplace approximations used for the prior and posterior is probably the way to go from the stochastic to the deterministic setting but this is not said clearly, while it is key to the overall analysis. This should be said clearly.

Approaching the contribution from the lens of a mix of theoretical results (from the PAC-Bayes theory, i.e. Catoni's bound) and empirical approaches, sets very high expectations from the editorial/reader viewpoint so as for the reader to know what is the purpose of each paragraph, each section, and the paper as a whole. The version of the paper as it is, shows a huge lack of such editorial discipline, and that is what is conveyed by the reviewers comments, that stop on various parts of the paper which seem hard to connect to each other.